# Epistemic Stance in Chinese L2 Spoken English: The Effect of Grade and Genre-Specific Questions

**Fang Xu and Rongping Cao ***

School of Foreign Languages, Beijing Forestry University, Beijing 100083, China
* Correspondence: caorongping@bjfu.edu.cn

**Abstract:** Zuczkowski et al.'s KUB model clarified three epistemic stances: Knowing/Certain, Not Knowing Whether and Believing/Uncertain, and Unknowing/Neither Certain nor Uncertain, according to the speakers' communicated information, and delineated three types of markers: macro-markers, micro-markers, and morphosyntactic markers. The model has seldom been applied to L2 instruction. To address this gap, the study examines the effect of grade and genre-specific questions on Chinese L2 speakers' choice of epistemic markers with reference to the model by analyzing the self-built corpus consisting of the oral data collected from two groups: Group One consisting of 20 sophomores and Group Two comprising 20 first-year graduate students. The participants were required to answer four genre-specific questions covering argumentation, description, narration, and exposition. The results show that the two group members use similar epistemic markers (EMs) for the Knowing/Certain and Not Knowing Whether and Believing /Uncertain positions but present a slight discrepancy in Unknowing/Neither Certain nor Uncertain stance-taking. The genre-based questions demonstrate a significant effect on the graduate speakers' use of the micro-markers and morphosyntactic markers for the Not Knowing Whether and Believing/Uncertain and the macro-markers and morphosyntactic markers for the Unknowing/Neither Certain nor Uncertain. It indicates that high-grade speakers are more sensitive to genre-based messages, though they use rather limited epistemic forms as low-grade speakers do. The findings suggest that English as a Second Language (ESL) oral instruction in China should be reformed and supplemented with diverse EMs to allow the speakers to take the epistemic stance they are comfortable with.

**Keywords:** epistemic stance; L2 spoken English; grade; genre-specific context

## 1. Introduction

Epistemic stance-taking is the core pragmatic skill in speaking and has a significant effect on the smooth flow of daily communication. Studies show that speakers tend to show more concern for marking their epistemic stance than marking attitudes or evaluations or expressing personal feelings and emotions in interactive communication (Biber et al. 1999; Thompson 2002). Pouromid (2021) concludes that speakers manage their epistemic stance to achieve intersubjectivity and maintain mutual understanding despite gaps in their linguistic repertoire. Therefore, the appropriate use of epistemic stance appears to be crucial and a challenge for L2 speakers in mastering speaking skills, allowing them to freely display their personalities and interact better with people.

Some scholars define epistemic stance or epistemic position as the speaker's commitment to the status of the communicated information, most commonly their assessment of its reliability (Kärkkäinen 2003; Biber 2004; Du Bois 2007; Englebretson 2007). Ochs (1996) incorporated sources of information (namely evidentiality) and the degree of certainty into the concept of epistemic stance. To substantiate Ochs (1996)'s perspective, Zuczkowski et al. (2014) further proposed the "KUB model", the acronym from "Knowing/Certain, Unknowing/Neither Certain Nor Uncertain, Believing/Uncertain". More recently, the epistemic stances were updated to "Knowing/Certain, Unknowing/Neither Certain Nor

Uncertain, Not Knowing Whether and Believing/Uncertain" so that evidentiality and epistemicity are likened to two sides of the same coin (Zuczkowski et al. 2017; Zuczkowski et al. 2021). According to Zuczkowski et al. (2021), epistemic stances are conveyed by lexical markers and are further classified into macro-markers and micro-markers, and morphosyntactic markers, pointing to three positions—Knowing/Certain position, Not Knowing Whether and Believing/Uncertain position, and Unknowing/Neither Certain nor Uncertain position—each having two sides, one evidential (source, access) denoting the left of the slash and the other epistemic (commitment), pointing to the right of the slash. The macro-marker is a general label and a hypernym, encompassing all the micro-markers that specify access to information or refer to a particular commitment to the truth of information in the here and now of communication. The morphosyntactic markers refer to syntactic structures communicating speakers' epistemic status.

## 2. Epistemic Stance in L2 Spoken Language Research

Recently, scholars have been exploring features in L2 spoken English and attempting to test elements affecting forms of epistemic stance.

Zhang and Sabet (2016) highlighted one typical epistemic lexical marker, I think (IT), and investigated its variation caused by different cultural backgrounds. The researchers drew on the large-scale naturally occurring spoken data produced by L1 and L2 speakers to investigate the elasticity of "IT", the most common epistemic marker. The result shows that L1 and L2 speakers have different preferences and focuses in using IT. L1 speakers are speaker-oriented and assertive, whereas L2 tend to take the middle ground between speaker-oriented/assertive with listener-oriented/less authoritative.

Some researchers have shown interest in the effect of task type. Gablasova et al. (2017) focused on two task types and tested the effect of the task on L2 speakers' stance-taking choice from three epistemic forms: adverbs, adjectives, and verb expressions. The outcome demonstrates that there is evidence that task and style can affect speakers' selection of epistemic markers, and the speakers used a specific pattern of stance-taking in the monologic task, as opposed to the interactive task where contextual factors are changing, and the two interlocutors interactively negotiate the epistemic stance. Their results were confirmed by the research conducted by Ou and Huang (2016) that investigated the effect of oral task type on learners' epistemic stance expression behavior and showed that the overall frequency of an epistemic stance maker in the conversation task is much higher than that in impromptu speech since the speakers use more epistemic stances to maintain the relationship with each other in a conversation task. However, both studies claim that the monologic tasks are under-researched.

More recently, researchers directed to test the effect of contextual factors and the interactive context demonstrated an effect on affecting the speakers' epistemic forms (Gablasova and Brezina 2015). Pérez-Paredes and Bueno-Alastuey (2019) explored the effect of contexts on epistemic adverbs used by native speakers (NSs) of English and non-native speakers (NNSs) of English across the same speaking task in the four datasets. They concluded that there is significant difference in the effect of contextual factors while different language groups of NNSs using epistemic adverbs.

Previous research has mainly focused on the lexical markers of the epistemic stance and tested the effect of cultural background, task types, and the contextual factors constructed by the interactive activities. There is little research on morphosyntactic epistemic forms and a lack of study on the effect of L2 oral instruction in monologic tasks on epistemic stance-taking. Moreover, the effect of language proficiency may be mitigated by learners involved in interaction, but in monologic tasks learners are left to their own resources, without being affected by their interlocutors (Robinson 2011).

According to Perry (1970), who focused on college students' epistemic growth, epistemic development continues long beyond childhood. Related studies have shown that senior college students' score on epistemic stance (Perry's Position) was significantly higher

than sophomores (King 1978; Widick et al. 1975; Meyer 1977; Clinchy et al. 1977; Chandler et al. 2002).

## 3. The KUB Model and Its Applications

The KUB model has been applied to investigate the authors' epistemic stance-taking in scientific written texts. Bongelli et al. (2012) detected the uncertain markers in 80 biomedical research articles, sourced from *The British Medical Journal* randomly sampled over a 168-year period. Results show that there is a trend that the uncertain markers gradually decreased in recall along with an increase in precision. Further research was conducted on a larger sample size and the outcomes delineate that the authors use lexical markers more frequently than morphosyntactic markers, and the writers have kept using uncertainty in an unaltered way and always occupy a smaller percentage with respect to certainty (Zuczkowski et al. 2016). The outcomes are confirmed by Bongelli et al. (2019)'s research that incorporated scientific popular articles from *Discover* 2013 for genre comparison, and statistics show that scientific and popular scientific journals demonstrate the gradual diminishing of uncertain markers over time, which are susceptible to the genre variant.

The KUB model has also been adopted to study questions in spontaneous conversations. Riccioni et al. (2018) investigated the relation between epistemic stance and four types of uncertain questions: alternative, polar interrogatives, tag, and declarative questions, in which polar and alternative questions are clarified as dubitative questions. Bongelli et al. (2018) found that wh-questions, on the one hand, and alternative and polar questions on the other come from two different epistemic positions: unknowing and uncertain, through the analysis of presuppositions, question design, social action, and preference organization in short fragments of Italian question–answer sequence.

Epistemic status and epistemic negotiation are other aspects of the KUB model that have been adopted to analyze literary types of discourse. Riccioni et al. (2014) investigated the role of epistemic negotiation and mitigation in the giving-advice sequence within informal troubles-talk and showed that epistemic status must be negotiated and shared by two interlocutors in order to have the advice accepted. Dorigato et al. (2015) analyzed dialogues in *Harry Potter* books and delineated how the characters negotiated, developed, and constructed their identities and how they evolved within the epistemic stance through the dialogue. Vincze et al. (2016) focused on the epistemic status in the presidential debate and found that the political candidates tended to employ ignorance-unmasking questions and multimodal negative evaluations of the opponent to boost their epistemic status and renegotiate their epistemic authority. Applications of the KUB model have not covered L2 instruction to date.

A literature search from databases including Web of Science, Google Scholar, ResearchGate, IEEE, Taylor & Francis, and Joster, using the keywords, "KUB model"+ "epistemic stance", and "L2 instruction", or "grade/age" indicates that applications of the KUB model have not covered L2 instruction discourses to date, nor investigated grade/age effect.

## 4. The Current Study

The current study attempts to address the aforementioned research gap by investigating the effect of grade and genre-specific questions on speakers' choices of EMs in oral ESL instruction. It seeks to answer the following research questions:

1.   Is there an effect of grade on L2 speakers' choice of epistemic stance?
2.   Is there a significant difference between epistemic markers employed by the sophomores and the first-year graduates across genre-specific questions?

Q1 explores whether there is a discrepancy in epistemic forms between the first-year graduate speakers and the sophomore speakers, such as their EM preferences, ranges, and diversity. The ranges depict the variation across the three types of EMs in administration, and the diversity points to the diverse items under each category. Q2 investigates whether their choice of EMs is susceptible to genre-specific questions.

## 5. Method

### 5.1. Participants

The present study selected second-year undergraduate and first-year graduate students as research subjects in an attempt to find the epistemic stance discrepancy across two grades. Forty students at Beijing Forestry University voluntarily participated in this program and were identified as two groups. Group One consists of 20 sophomores from diverse colleges comprising Forestry, Soil, and Water Conservation, Landscape Architecture, Biological Sciences and Biotechnology, Economics and Management, and Technology, comprising 7 males and 13 females with an average age of 20. They all passed College English Test Band 4 (CET-4) and have received two-year oral English training in college. Group Two consists of 20 first-year graduate students, from colleges of Forestry, Biological Sciences and Biotechnology, Technology, and Foreign Languages, involving 6 males and 14 females with an average age of 23, who passed College English Test Band 6 (CET-6) and have received oral English training for four years. The two group members' mother tongue is Chinese, and they have never learned a third language. CET-4 and CET-6 are the standard tests for gauging college students' language proficiency administrated by the Ministry of Education of the People's Republic of China (MoEPRC). The basic information of the participants is summarized in Table 1:

**Table 1.** The demographic information of the two groups.

| Group | Gender | Average Age | Level | College |
|-------|--------|-------------|-------|---------|
| One | 7M&13F | 20 | CET-4 | Forestry/Soil and Water Conservation/Landscape Architecture/Biological Sciences and Biotechnology, Economics and Management/Technology. |
| Two | 6M&14F | 23 | CET-6 | Forestry/Biological Sciences and Biotechnology/Technology/Foreign Languages |

### 5.2. Data Collection

#### 5.2.1. Collecting Recordings

The participants are required to record their answers to the four questions on two topics: Marriage and Love and Embarrassing Experience. They are supposed to upload to the WeChat platform two extracts, each of which is no longer than 3 min. Each topic involves two related questions, encompassing argumentative, descriptive, narrative, and expositive messages. The subject is supposed to submit a persuasive response plus a descriptive response for the topic, Marriage and Love, and a narrative response plus an expositive response for the topic, Embarrassing Experience. To collect the participants' spontaneous speeches, the participants were required to submit their answers in 3 min for each question, but they still had 30 s for preparation before starting to record. The recordings will be submitted automatically when the time is up and resubmission is unacceptable. The topics and the contained questions are presented below:

Marriage and Love

- What do you think of cyber love? (Your opinion plus reasons)
- Please describe the qualities of your expected life companion.

Embarrassing Experience

- What is your most embarrassing experience? (Please narrate your experience in detail)
- How can you overcome social phobia and be more outgoing?

#### 5.2.2. Coding the Epistemic Markers

According to Zuczkowski et al. (2021)'s KUB model, the most typical marker for communicating a piece of information from the Knowing/Certain position is the plain

declarative sentence in the indicative mood (either present, past, or future) without any lexical marker of epistemicity or evidentiality (Lyons 1968; Aijmer 1980). For instance, "Someone was dancing to the music." The sentence does not contain any lexicons indicating the source of information or the speaker's commitment to the truth of the communicated information. Providing that the declarative sentences include lexical evidential or epistemic markers, such as "I know/I am certain", the general type, adaptable to many syntactic structures and contexts, and "I remember/no doubt", the specific type, subject to the limited contexts, they will be identified as macro-markers or micro-markers, respectively. The former is described by Fordyce (2014) as "easy forms", and the latter as "difficult forms". The EMs are summarized in Table 2.

**Table 2.** The Knowing/Certain stance and its related EMs.

| Macro-Markers | Micro-Markers | Morphosyntactic Markers |
|---|---|---|
| Knowing/I know . . . Certain/I'm certain . . . | I remember . . . /I hear . . . /I see . . . No doubt . . . /surly/certainly/of course/without doubt | The plain declarative sentence without lexical markers |

The EMs were summarized from Zuczkowski et al. (2021)'s "KUB"model.

The uncertain position indicates that the speakers' commitment to the truth is at a minimum level, and it turns out to be an epistemic continuum ranging from two opposing poles of not knowing whether something is true or not to believing that something is true or false. The uncertain markers can be grouped into six categories: verbs, modal verbs, non-verbs, uncertain questions, if-clauses, and epistemic future. Verbs, modal verbs, and non-verbs belong to lexical categories whereas uncertain questions, if, and epistemic future are classified as morphosyntactic markers. The uncertain position highlights the modal verbs in the simple present tense, communicating possibility or uncertainty, such as can, may, and the modal verbs in the conditional mood, expressing uncertainty, including could, might, and should. Non-verbs involve adjectives, adverbs, and nouns, which convey uncertainty and doubts, and expressions concerned with personal opinions. Polar interrogatives, tags, and declarative questions are considered as conveying the speakers' uncertainty. If-clauses are counted in uncertain markers except for the zero conditionals (Zuczkowski et al. 2021)[1]. All other forms of if-clauses are markers of uncertainty. For example, "If I have money, I will buy a boat". "If I had had enough money, I would have bought that boat ". The epistemic future is realized by the epistemic or conjectural uses of "will". For example, "(The doorbell is ringing.) That will be the postman" can be paraphrased as "I (now) hear the doorbell ringing and make an assumption (Uncertainty) that the postman is at the door", based on prior knowledge (concerning, for example, the fact that the postman usually arrives at that time and, if he has to deliver mail, he rings the bell) (Palmer 2001). Table 3 summarized the markers for the Not Knowing Whether and Believing/Uncertain stance.

**Table 3.** The Not Knowing Whether and Believing/Uncertain stance and its related EMs.

| Macro-Markers | Micro-Markers | Morphosyntactic Markers |
|---|---|---|
| I don't know whether . . . /I believe . . . I'm uncertain . . . /I'm not certain . . . | uncertain/possible/probable/ supposed/assumed/believed/ doubted/suspected/I think/I suppose/I doubt/I guess/in my opinion/according to me/as far as I am concerned | modal verbs in conditional and subjunctive moods If clauses Epistemic future |

The EMs were summarized from Zuczkowski et al. (2021)'s "KUB" model.

The unknowing position communicates the speakers' absence of information. Consequently, the speaker cannot express his certainty or uncertainty due to the information void. The typical marker is "I don't know" encompassing the micro-markers denoting

evidentiality, such as "I don't remember", "I don't see" and "I don't hear", and some adjectives, such as "incomprehensible"," mysterious", "obscure", and so on. The speaker asks literary interrogatives for seeking information, implying the speaker's lack of information. However, rhetorical questions or question tags should be excluded since those questions are expected to act as an effective tool wielded by the speakers to make their words impactful or concise, apart from communicating their epistemic status. For instance, "It is a sunny day, isn't it?" The speaker may simply seek agreement/confirmation or merely get in touch with others, instead of asking for information. The markers are listed in Table 4.

**Table 4.** The Unknowing/Neither Certain nor Uncertain stance and its related EMs.

| Macro-Markers | Micro-Markers | Morphosyntactic Markers |
|---|---|---|
| Unknowing/I don't know … | I don't remember/I don't see/I don't hear/incomprehensible/ mysterious/obscure/mystery/ secret/ambiguity | "Literal" interrogatives (i.e., excluding rhetorical questions, question tags, etc.) |

The EMs were summarized from Zuczkowski et al. (2021)'s "KUB" model.

### 5.3. Data Analysis

Altogether, 120 tracks equaling 3 h, 54 min, and 26 s of voice recordings are collected. Each participant's recording times vary between 5 min and 6 min. The voice recordings were transcribed through the online platform: https://www.iflyrec.com (accessed on 10 August 2022) and saved as separate plain text files. The authors listened to the recordings, checked the words, and made an agreement on revision. Thereby, according to the group, the author built two datasets: the Group One dataset encompassing a total of 36,607 tokens, and the group two dataset consisting of a total of 46,533 tokens.

The authors first conducted literary concordance using Nvivo to identify the listed lexical markers and counted the frequencies of each marker. A total of 12 markers were identified for the Knowing/Certain position, 17 markers for Not Knowing Whether or Believing/Uncertain, and 11 markers for Unknowing/Neither Certain nor Uncertain. After that, the authors did the manual analysis to capture the epistemic forms that cannot be identified by the tool. The authors separately identified the morphosyntactic markers and made decisions for inclusion based on the KUB model, sought agreement, and combined the files to check the average Kappa value, which is 0.95, indicating high reliability. For morphosyntactic markers, one independent sentence is counted as a single unit for statistical analysis. The item-level descriptive statistics of the two datasets are presented in Tables 5 and 6.

**Table 5.** The item-level descriptive statistics of the Group One dataset.

| | N | Mode | Std. Error | Std. Deviation |
|---|---|---|---|---|
| Marker | 421 | 3 [1] | 0.027 | 0.546 |
| Genre type | 421 | 3 [2] | 0.052 | 1.063 |
| Epistemic stance | 421 | 1 [3] | 0.025 | 0.523 |
| Valid N (listwise) | 421 | | | |

[1] 3 = morphosyntactic marker. [2] 3 = argumentation. [3] 1 = Knowing/Certain stance.

**Table 6.** The item-level descriptive statistics of the Group Two dataset.

| | N | Mode | Std. Error | Std. Deviation |
|---|---|---|---|---|
| Marker | 737 | 3 [1] | 0.018 | 0.494 |
| Genre type | 737 | 1 [2] | 0.040 | 1.093 |
| Epistemic stance | 737 | 1 [3] | 0.032 | 0.868 |
| Valid N (listwise) | 737 | | | |

[1] 3 = morphosyntactic marker. [2] 1 = narration. [3] 1 = Knowing/Certain stance.

The frequencies of the two groups' three types of EMs to the four types of questions were counted under each epistemic stance, supplemented with qualitative analyses of the transcribed texts to capture the two group members' ranges, preferences, and diversity of EMs across the four genre-specific questions, pertaining to Q1. For Q2, SPSS is adopted to do a chi-square test to examine the effect of genre-based questions on the two groups' EMs. The authors also used the partitions of $X^2$ method to further detect which types of EMs adopted by the speakers were more susceptible to genre-specific messages.

## 6. Results and Discussion

The distribution of each group's EMs of the three epistemic stances to the four types of questions is present in the bar charts, followed by examples extracted from the data.

The chi-square values showed the genre-specific questions' effect on the ESL speakers' choice of types of EMs for communicating the epistemic position. The data would be further analyzed leveraging the partitions of the $X^2$ method in the case of a significant effect detected.

The outcomes would be expounded on in the following sections. Group One members were tokened with "US", and Group Two members with "GS".

### 6.1. The Distribution of EMs Generated by the Three Epistemic Positions

Figures 1 and 2 demonstrate the two group members' EM distributions for knowing/certain stance-taking. Overall, the two group members are inclined to use morphosyntactic markers for the knowing/certain position, especially in narrations, followed by argumentations, as shown in Figures 1 and 2. In contrast, the percentages of the lexical markers account for a relatively small proportion across the four types of questions.

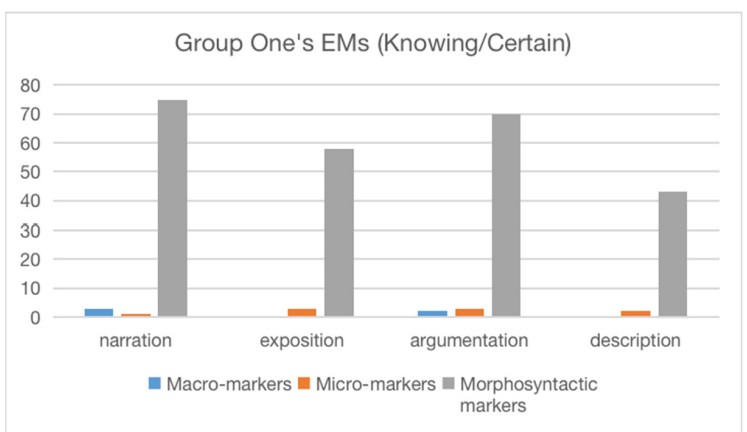

**Figure 1.** The bar chart of Group One speakers' EMs (Knowing/Certain).

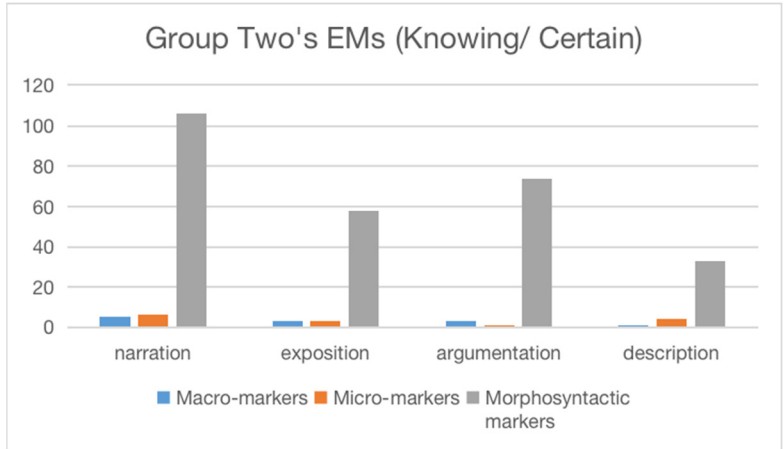

**Figure 2.** The bar chart of Group Two speakers' EMs (Knowing/Certain).

The typical morphosyntactic marker expresses that the knowing position is the declarative sentence: the default type of the independent clause retaining the declarative force of the statement (Biber et al. 2003). It occupies a large area in narrations because the plain statements are suitable for relating a series of events. Moreover, the argumentation is more convincing in the absence of subjective lexical expressions, whereas in exposition or description, declarative sentences occupy a relatively small percentage since the expected information is related to personal opinions, which are more likely to go with lexical markers indicating mental thoughts. The data provide evidence that declarative sentences are more effective for communicating information, especially for relating facts or evidence as opposed to opinions. For example,

- US6: And one of my most classic embarrassing experiences is when I'm on class.
- GS6: In my second year, in high school, I once liked a boy who was my senior, he was a sports student with special talent.
- GS6: We can understand a person besides information and some family situations through the network, but to rarely understand the habits of this person,
- GS14: Some people claim that online love is very romantic and exciting. It's a amazing things because the internet brings two strange persons far away from each other together.

Some grammatical mistakes can be seen from the examples, such as "on class", "besides information", "but to rarely understand", and "a amazing", suggesting that even high-grade speakers may utter ungrammatical sentences in spontaneous speeches, and opt for the plain declarative sentences for telling stories or providing evidence, similar to the low-grade speakers. The results indicate that the speakers' cognitive mechanism of communicating information outperforms the monitoring nerves wired in their brains. The ESL oral training in China contributes a little to develop the ESL speakers' vocabulary complexity in conveying the knowing stance. As a result, they are apt to use the standardized forms to communicate their knowing position, albeit with different messages.

Group One prefers to use micro-markers across the four genre-based questions when expressing the Not Knowing Whether and Believing/Uncertain stance, as shown in Figure 3. The same feature can be detected in the Group Two members' argumentative and descriptive discourses, with morphosyntactic markers claiming a dominant proportion in narration and exposition, contrastive to the Group One counterparts, as demonstrated in Figure 4.

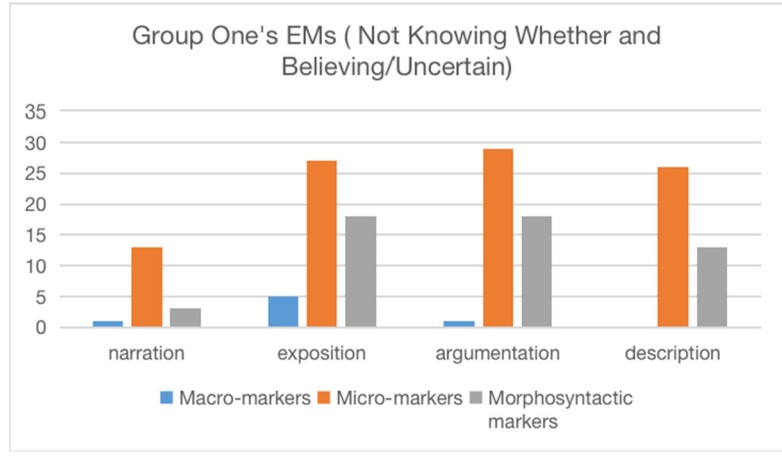

**Figure 3.** The bar chart of Group One speakers' EMs (Not Knowing Whether and Believing/Uncertain).

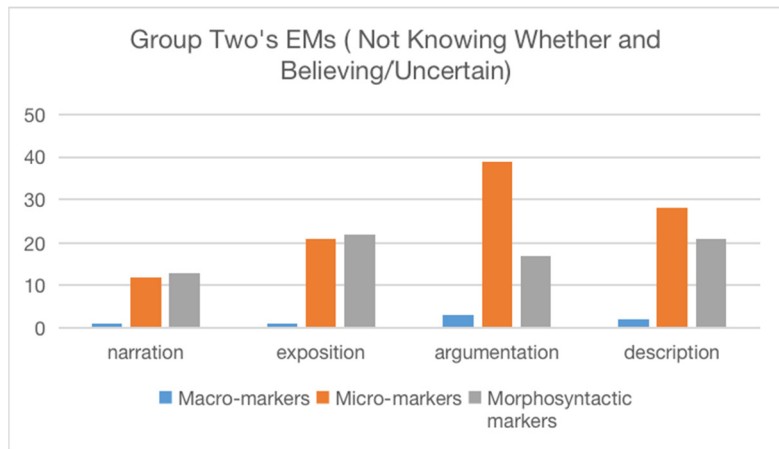

**Figure 4.** The bar chart of Group Two speakers' EMs (Not Knowing Whether and Believing/Uncertain).

Similar features of the two groups can also be detected in the diversity and ranges of micro-markers, The typical micro-markers are the first-person-perspective expressions, such as "I think", "in my opinion", "from my perspective", and "as far as I'm concerned" since Chinese NNSs tend to literally translate chunks in Chinese (Xu and Xu 2007). Huang (2014) found that "I think" is likely to co-occur with hesitation markers, pauses and restarts, personal opinions and evaluation, factual information, concluding remarks, and questions. The micro-markers may co-occur with different structures across the four stylistic questions.

For the typical micro-markers, the first-person perspective expressions primarily co-occur with personal opinions in argumentation. A complete argument consists of three building blocks: claims, warrants, and reasons (Zhang 2011; Bennett 2015; Bowell et al. 2019). The speakers use "micro-markers plus personal opinions" more often to state their claims, employ "because plus that-clause" for reasons, and are inclined to adopt implicit warrants. Sometimes, "that" is added due to personal styles, or the whole phrase forms a lexical trunk implemented in the speaker's head. Some examples listed below:

- GS11: <u>I think</u> cyber love is a very scared thing.
- GS5: <u>As far as I am concerned</u>, it is an easy and fast way for people to make new people by the internet.
- GS9: <u>I think that</u> online dating is unreliable, because in the virtual species of the internet, everyone can disguise.

It is the same with description, where the micro-markers precede personal opinions. However, in most cases, the micro-markers may occur in the rear or with the hesitation markers, "er" and "um", indicating the speaker is searching for content information or appropriate lexical expressions. For example,

- GS10: <u>I think</u> first, first and the most important and <u>I think</u> he will very handsome. Yes. And <u>I think um</u> if if I see a boy who is very handsome, I will have the good emotion.
- US 10: <u>I think</u> this question is a very complex question that I can't give the clear answers about it. <u>In my opinion</u>, <u>I think</u> the life company must have the same or the similar attitudes about the life and the future.

In the second case, the co-occurrence of "in my opinion" and "I think" demonstrates their function as filler words, suggesting the mental process, as NNSs usually need more time to express their meaning in a foreign language. The two group speakers tend to mitigate using the micro-markers and the graduate speakers are more likely to use the formulaic structure, "first, second, third . . . " when describing their expected companions' qualities. For example,

- US15: My standard for choosing a partner is that he need to be a good character, motivated tyrant, and love me.

- GS12: <u>First of all</u>, my life company must be a person with three views of integrity and confirm to my concepts, which can guarantee a effective communication. <u>Secondly</u>, it is essential for him to be self-motivated and waiting to fight with me for the <u>future life</u>. Finally, his appearance should miss my synthetic needs.

For the exposition, the two group speakers use the micro-markers to give suggestions on overcoming social phobia, closely concerned with tackling a problem. The first-person perspective markers act as softeners to avoid being assertive and sound more acceptable to listeners. The morphosyntactic marker, "if-clause", is the optimal option for the speakers to offer suggestions. For example,

- GS2: <u>I think that if</u> you want to overcome social phobia, you must first overcome psychological barriers, don't be afraid to chat to death.
- US10: <u>if</u> you're embraced about communicating in the real world, <u>I think</u> it is a better choice for you to make some friends online

In summary, the EMs employed by the two grades for conveying the Not Knowing Whether and Believing/Uncertain are of high variability, tightly correlated with the speakers' subjective mental activities but still confined in a certain range. The group members tend to take this stance to communicate pragmatic meanings, apart from revealing their epistemic status.

The Unknowing/Neither Certain nor Uncertain stance represents relatively small proportions in the two groups. The significant discrepancy can be detected in Figures 5 and 6. Figure 5 shows that the speakers prefer to use macro-markers for explaining their lack of knowledge, as opposed to the Not Knowing Whether and Believing/Uncertain position, where the micro-markers dominate the speeches. The similarity can be captured in argumentation in which the two group members are inclined to use macro-markers to support their claims in argumentation, especially for the opposite opinions. For example,

- GS8: I think the relationship is about the cyber love is not very good, because we can't, because <u>we can't know about</u> the guys we met in. We met in the internet. <u>We don't know</u> his characteristics, and we don't know a lot of things about it.
- GS1: First of all, cyber love is very unsafe on the internet. <u>We don't know</u> what the other person looks like. <u>Don't know</u> the Information.
- US14: Instead of spending a lot of time talking to someone <u>you don't know</u>, focus on learning, practicing, and improving yourself.

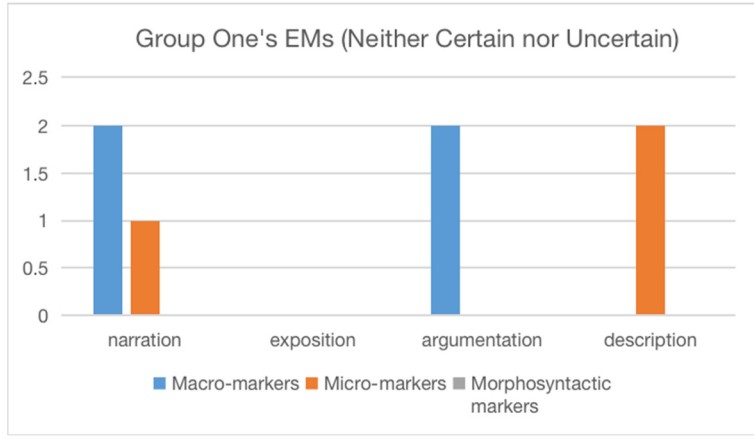

**Figure 5.** The bar chart of group one speakers' EMs (Unknowing/Neither Certain nor Uncertain).

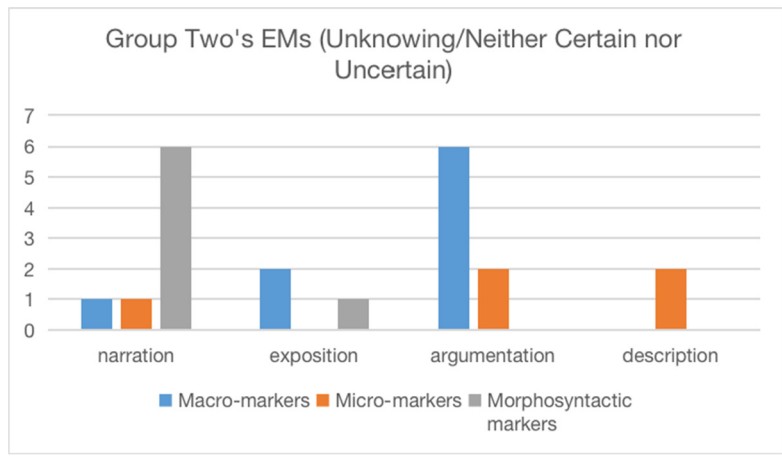

**Figure 6.** The bar chart of Group Two speakers' EMs (Unknowing/Neither Certain nor Uncertain).

When relating their embarrassing experiences, the Group Two speakers tend to ask questions to vividly depict the mental process, whereas the undergraduate speakers may generally reproduce their thoughts. For example,

- GS14: And I couldn't understand the quiz. <u>What is</u> the <u>question</u>? So I asked her, again, <u>what the question is</u>?
- GS7: There are a lot of thoughts disturbing me. <u>Am I performing good? Can I do my best? How do others think of me?</u>
- US7: Well, it was so embarrassing, and <u>I even didn't know</u> what to do.

### 6.2. The Effect of Genre-Based Contexts on EMs across the Three Epistemic Positions

The chi-square test is adopted to check the variation of EMs across the four questions, categorized by the three epistemic positions.

Via the chi-square test, Table 7 shows that the calculated *p*-values of each group are 0.496 > 0.05 (Group One) and 0.519 > 0.05 (Group Two), respectively, indicating that there is no significant discrepancy for the two groups in using EMs across the four questions to take the Knowing/Certain stance. That means both groups apt to use similar EMs across the genre-based questions.

**Table 7.** The chi-square test of the EMs (Knowing/Certain).

| | Group | Value | Df | Asymptotic Significance (Two-Sided) |
|---|---|---|---|---|
| Group One | Pearson chi-square | 5.379 | 6 | 0.496 |
| | likelihood ratio | 7.372 | 6 | 0.288 |
| | Linear-by-linear association | 0.160 | 1 | 0.689 |
| | N of valid cases | 260 | | |
| Group Two | Pearson chi-square | 5.192 | 6 | 0.519 |
| | likelihood ratio | 5.301 | 6 | 0.506 |
| | Linear-by-linear association | 0.045 | 1 | 0.831 |
| | N of valid cases | 297 | | |
| Total | Pearson chi-square | 4.918 | 6 | 0.554 |
| | likelihood ratio | 4.917 | 6 | 0.554 |
| | Linear-by-linear association | 0.312 | 1 | 0.557 |
| | N of valid cases | 557 | | |

Table 8 shows that the chi-square value of group one is 8.585 ($p = 0.198 > 0.05$), whereas the chi-square value of group two is 42.065 ($p = 0.000 < 0.01$), suggesting a great significance detected in the co-relation of EMs with the questions. The results delineate that the Group Two speakers' use of EMs explaining the Not Knowing Whether or Believing/Uncertain

position varies across the genre-based questions. In order to figure out the specific EMs, the authors employed the partitions of the $X^2$ method for further calculations, and the results are further detected and analyzed.

**Table 8.** The chi-square test of the EMs (Not Knowing Whether and Believing/Uncertain).

|  | Group | Value | Df | Asymptotic Significance (Two-Sided) |
|---|---|---|---|---|
| Group One | Pearson chi-square | 8.585 | 6 | 0.198 |
| | likelihood ratio | 9.918 | 6 | 0.128 |
| | Linear-by-linear association | 1.942 | 1 | 0.163 |
| | N of valid cases | 154 | | |
| Group Two | Pearson chi-square | 42.065 | 6 | 0.000 |
| | likelihood ratio | 51.036 | 6 | 0.000 |
| | Linear-by-linear association | 19.625 | 1 | 0.000 |
| | N of valid cases | 147 | | |
| Total | Pearson chi-square | 22.732 | 6 | 0.001 |
| | likelihood ratio | 23.382 | 6 | 0.001 |
| | Linear-by-linear association | 7.551 | 1 | 0.006 |
| | N of valid cases | 301 | | |

Table 9 presents the Group One Pearson chi-square value: $X^2 = 4.278$, $p = 0.118 > 0.05$, and the Group Two Pearson chi-square value: $X^2 = 18.428$, $p = 0.005 < 0.01$. The $p$-value is an indicator for a significant level, and the significance is detected in Group Two, demonstrating that EMs communicating the Unknowing/Neither Certain nor Uncertain position vary across the four questions, or the effect of the genre-based questions is significant on Group Two members' options for EMs. The partitions of the $X^2$ method for multiple comparisons are applied to obtain the variance further.

As shown in Table 10, the significance can be detected in the pairwise comparison between the micro-markers and morphosyntactic markers ($p = 0.000 < 0.0125$). It manifests that the Group Two speakers' use of the micro-markers and morphosyntactic markers for taking the Not Knowing whether and Believing/Uncertain positions are susceptible to the inbuilt genre-specific information. The effect of genre-based questions on two types of EMs is as follows: micro-markers and morphosyntactic markers are significant in the graduate speakers' expressions, pointing to the Not Knowing Whether and Believing/Uncertain position.

**Table 9.** The chi-square test of the EMs (Unknowing/Neither Certain nor Uncertain).

|  | Group | Value | Df | Asymptotic Significance (Two-Sided) |
|---|---|---|---|---|
| Group One | Pearson chi-square | 4.278 | 2 | 0.118 |
| | likelihood ratio | 5.742 | 2 | 0.057 |
| | Linear-by-linear association | 0.878 | 1 | 0.349 |
| | N of valid cases | 7 | | |
| Group Two | Pearson chi-square | 18.428 | 6 | 0.005 |
| | likelihood ratio | 20.396 | 6 | 0.002 |
| | Linear-by-linear association | 6.139 | 1 | 0.013 |
| | N of valid cases | 21 | | |
| Total | Pearson chi-square | 21.516 | 6 | 0.001 |
| | likelihood ratio | 23.686 | 6 | 0.001 |
| | Linear-by-linear association | 3.694 | 1 | 0.055 |
| | N of valid cases | 28 | | |

**Table 10.** The multiple comparisons of the EMs (Not Knowing/Uncertain) generated by Group Two.

| Comparative Markers | Group | X² | *p*-Value |
|---|---|---|---|
| Macro-markers Micro-markers | Two | 7.620 | 0.055 |
| Macro-markers Morphosyntactic markers | Two | 7.696 | 0.053 |
| Micro-markers Morphosyntactic markers | Two | 38.845 | 0.000 |

Table 11 shows that a great significance resides in the comparisons of macro-markers and morphosyntactic markers ($p = 0.007 < 0.0125$). It demonstrates that when conveying the Unknowing/Neither Certain nor Uncertain position, the graduate speakers are sensitive to the inbuilt genre-specific messages when choosing the macro-markers and the morphosyntactic markers. For example, they use "don't know" to support the antagonistic claims and questions for recounting stories, as opposed to the undergraduate speakers who make no distinction and overuse the macro-markers regardless of the genre types.

**Table 11.** The multiple comparisons of the EMs (Unknowing/Neither Certain nor Uncertain) generated by Group Two.

| Comparative Markers | Group | X² | *p*-Value |
|---|---|---|---|
| Macro-markers Micro-markers | Two | 5.289 | 0.152 |
| Macro-markers Morphosyntactic markers | 5.301 | 9.808 | 0.007 |
| Micro-markers Morphosyntactic markers | Two | 8.473 | 0.037 |

Via observation, pertaining to Q1, the two grades use rather limited and similar EMs, especially when expressing the Not Knowing Whether and Believing/Uncertain stance, in which the first-person perspective micro-markers dominate the expressions. However, the discrepancy is found in expressing the Unknowing/Neither Certain nor Uncertain stance where the graduate speakers tend to use questions to recount stories, and the undergraduate speakers prefer to resort to the macro-markers for stating their psychological activities.

As for Q2, the effect of the genre-based questions is captured in Group Two's use of the two epistemic stances: the Not Knowing Whether and Believing/Uncertain stance and the Unknowing/Neither Certain nor Uncertain position. The high-grade speakers show sensitivity to the genre-based questions in terms of the micro-markers and the morphosyntactic markers for the Not Knowing Whether and Believing/Uncertain epistemic stance-taking, and the macro-markers and the morphosyntactic markers for the Unknowing/Neither Certain nor Uncertain epistemic position. In contrast, low-grade speakers prefer to use stable EMs regardless of the inbuilt genre information.

## 7. Conclusions

The article examined the effect of grade and genre-based questions on L2 speakers' EMs, based on the "KUB" model that clarifies three types of EMs—lexical macro-markers, micro-markers, and morphosyntactic markers, pointing to three epistemic positions: the Knowing/Certain stance, the Not Knowing Whether and Believing/Uncertain stance, and the Unknowing/Neither Certain nor Uncertain stance. The sophomores and the first-year graduate speakers recorded their answers to the four monologic speaking tasks embedded with the four types of genre-specific information: narration, description, exposition, and argumentation.

The high-grade speakers are more sensitive to the genre-specific messages in the questions but adopt similar epistemic forms as the low-grade speakers. This indicates that oral English pedagogy and English proficiency tests in China primarily aiming to cultivate students' awareness of implicit messages and develop students' epistemic expressions retreat into the second position. The spoken pedagogical design contributes to raising Chinese L2 speakers' genre awareness but proves insufficient in expanding the Chinese L2 speakers' epistemic forms. Consequently, the speakers may lack the ability to exactly express their epistemic status or properly interact with foreigners in intercultural communication. It is necessary to reform the traditional pedagogical design by introducing more epistemic forms to instruct Chinese L2 speakers, thus allowing them to better communicate their thoughts and take the stance they are comfortable with. However, this study is limited in that it was conducted on a small sample size, and future research will have to collect more data to dig deeper into the effect of ESL instruction in China on students' epistemic stance-taking. In addition, the four questions were insufficient in that they offered rather limited contexts: marriage and love, and embarrassing experiences. More contexts should be provided in order to explore whether the effect of genre-specific messages is regulated by the contexts.

Future research might focus on Chinese L2 learners with the same level of English language proficiency and make comparisons of their epistemic stance-taking in genre-specific texts in the same way as in verbal English and might perceive whether there are discrepancies in EMs regarding written texts and verbal expressions. If any, what is the degree of EMs variance? Apart from that, future studies can investigate whether L2 speakers take different epistemic stances to genre-specific questions under diverse contexts. The monologic tasks deserve to be further researched.

**Author Contributions:** Conceptualization, F.X. and R.C.; methodology, F.X.; software, F.X.; validation, F.X.; formal analysis, R.C.; investigation, F.X.; resources, F.X.; data curation, F.X.; writing—original draft preparation, F.X.; writing—review and editing, F.X.; visualization, F.X.; supervision, R.C.; project administration, R.C.; funding acquisition, R.C. All authors have read and agreed to the published version of the manuscript.

**Funding:** This research was funded by Beijing Forestry University education and research project, grant number #BJFU2019JY074.

**Institutional Review Board Statement:** Ethical review and approval were waived for this study due to the fact that this research does not require approval by the Human Sciences Ethics Committee of the Beijing Forestry University.

**Informed Consent Statement:** Informed consent was obtained from all subjects involved in the study.

**Data Availability Statement:** Data are available on request due to restrictions.

**Acknowledgments:** The authors would like to thank the students who participated in the research during the data collection process. The authors also thank the reviewers or editors for their useful comments and suggestions.

**Conflicts of Interest:** The authors declare no conflict of interest.

## Notes

[1]  Zero conditional (Zuczkowski et al. 2021): "if" is accompanied by the simple present in the conditional clause as well as simple present in the main clause. For instance, the sentence, "If the weather is fine, I usually go for a hike" is excluded since "if" can be replaced by "when" or "every time", which communicates certainty.

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
