# Peer review of "Epistemic Stance in Chinese L2 Spoken English: The Effect of Grade and Genre-Specific Questions"

_languages, doi:10.3390/languages8010015_

Round 1

Reviewer 1 Report

The manuscript investigates an interesting area within the study of epistemic, but it needs improvement in a number of aspects before getting published. 

General Issues:

1. Th literature review is too limited

2. In the methodology section, it is unclear what has happened to the experimental group. What kind of treatment did they receive?

3. What does justify a comparison between graduate and undergraduate students? Why does this variable matter? Do all graduate students have a common feature? Why is this variable relevant?

4. The conclusion section is only a reiteration of what has been said before. What can the authors conclude from the analysis of their data? What is the significance of the conclusion they draw?

Specific issues:

Line 66: What is IT? Acronyms need to be described the first time they are used.

Line 85: Why is "The" capitalized? 

Line 93: What is "the research"?

Line 111: What does grade refer to hear? Needs explanation early in the manuscript.

Line 139: What is naturalness? 

Lines 188 and 209: What is the Error thing?!

Author Response

We appreciate your comments and have revised the manuscript accordingly. We checked the spelling and made extensive changes:

  1. We enriched the literature review and rearranged the part into two sections: Epistemic Stance in L2 Spoken English and the KUB model and its Application. The first section reviewed the research status of the epistemic stance in L2 oral English and pointed out the limitation: uncovering the morphosyntactic markers of epistemic stance. The second section recounts the primary domains to which the “KUB” model has been applied, thereby addressing the gap that the model has never been adopted to investigate EFL instruction.
  2. We replaced the expression “the experimental group”, and “the control group” with “Group one”, and “Group two” since the two groups are equal. Group one consists of sophomores who received a two-year English oral training in college and passed College English Test Band Four. Group two comprises 20 graduate year one who has been orally trained for four years in college and passed College English Test Band Six.
  3. The characteristics of the two groups can reflect the effect of Oral English pedagogy and the purpose of foreign language proficiency tests in China.
  4. The conclusion part is added with an explanation of the statistics. The high-grade students are more sensitive to the in-built stylistic messages, indicating that the Oral English pedagogy and proficiency tests in China weigh on enhancing students’ awareness of the format instead of amplifying the students’ epistemic forms. China should reform English oral training by supplementing the pedagogical design with more epistemic forms, thus allowing the students to take the stance they are comfortable with.

Specific issues:

  1. IT is the abbreviated form of “I think”. We replaced IT with “I think”, for the whole phrase should be provided the first time it appears.
  2. We did not purposely capitalize “the” in Line 85, which may be a miss-spelling.
  3. The grade refers to the level of study in college, such as fisherman, sophomore...
  4. “Naturalness” refers to the participants’ spontaneous speeches without former instruction, and we decided to remove the word since it sounds improper.
  5. The errors may be caused by the missing codes, and they were tokens of figures.

We appreciate the valuable advice and will upload the revised manuscript soon.

Reviewer 2 Report

The article proposes an interesting application of Zuczkowski et al.'s KUB model in the analysis of a self-built corpus in which participants (Chinese L2 English speakers) answer four questions covering two different topics and involving the use of practices such as argumentation, description, narration and exposition. The analysis focuses on the use of epistemic markers and the results (as might be expected) reveal differences related to the subjects' level of expertise in L2.

One of the main novelties of the article concerns, in my opinion, the application of the KUB model to a corpus of monological speech elicited by researchers with specific purposes and produced by subjects speaking English as L2. In fact, the KUB model has so far been applied to the analysis of scientific texts (Zuczkowski et al. 2016; Bongelli et al. 2019; Omero et al. 2020; Riccioni et al. 2021), of spontaneous conversations in Italian and English (Riccioni et al. 2014, 2018; Zuczkowski et al. 2017, 2021; Bongelli et al. 2018, 2020a, 2020b), of literary dialogues (Philip et al. 2013; Dorigato et al. 2015), as well as in survey-based studies (Riccioni et al. 2022).I think that this research represents a valuable contribution to the field. However, I think that the manuscript in its current form is not suitable for publication and requires major revision, since there are some important issues that should be addressed.

Major points:

-       There is a mistake constantly repeated throughout the text of the article: the Uncertainty position in the KUB model is not ‘not knowing/uncertain’ but ‘not knowing whether/uncertain’. The simple ‘not knowing’ indicates another position, that of Unknowing (U). I suggest correcting with an automatic search for the expression, given the number of occurrences (and the importance of the concept)

-       “lexical macro-markers” and “lexical micro-markers”, as far as I know, do not belong to the terminology used by Zuczkowski et al. If it is a re-elaboration made by the authors, this point should be specified and illustrated more clearly.

-       The section Method has several points that need to be clarified and argued:

-     More information should be provided in the description of the participants (at least basic socio-demographic characteristics such as gender and age).

-       What do the authors refer to when they mention the “newest version of the KUB model” (line 151)? To one of their own reworkings for which they do not provide the reference because deleted for review? This should be clarified because the reader might imagine that the reference is to publications by Zuczkowski et al. (not cited here). In short, this passage (150-154) seems rather unclear and should be disambiguated, also because it will become crucial in the subsequent analysis. The categories of “macro-markers” and “micro-markers” themselves should be better explained and illustrated.

-       Lines 167-170: this passage needs to be clarified, since it implies important methodological issues and raises several questions: Nvivo, like other data processing tools, does not capture epistemic markers on its own, but has to be ‘trained’ by the researchers for automatic detection: this step should be described. Which and how many markers have been selected for automatic detection? And how? How did the authors proceed? This also serves to clarify the part of the subsequent manual analysis. But also here it should be clarified how the latter was carried out: did the authors analyse the transcripts together? Or did they analyse them separately in order to have a comparison and seek an agreement? Or did only one of the authors do this? 

-       It is not clear what data is used for the calculation of significance: words, markers (each counted as a single unit)? Was the calculation performed by normalised data? How does the total number of transcribed words relate to the number of markers

Summarising: the whole method section needs to be clarified and deepened  

-       I think the part concerning statistical analysis needs to be reviewed by an expert in the field (I am no such person).

-       The Conclusions are overly concise. There is a lack of discussion of the results considering the literature. No mention is made about the limitations of this study (which would be numerous) as well as its possible future developments. The impact of the research is only minimally presented and should be further investigated.

Other details:

-       The English language (L2) should be mentioned in the text of the abstract to avoid confusion for the reader

-       Line 32: Speakers, lower case

-       Line 66: “I think” should be made explicit

-       Line 81: add ‘s’ (plural) to ‘marker’

-       Line 88-89: “the four dataset”: should ‘the’ be removed?

-       Pay attention to the sections’ numbering (3 for The Current Study and Method)

-       Section 3: It should be made clear that L2 is the English language

-       Lines 111-112: it could be a reading aid to use Q1 and Q2 instead of 1 and 2

-       Line 112: “the two groups” is an anticipation which presupposes that the reader has read the abstract (while the participants are described in the next section)

-       Line 162: three à 3

-       Line 165: “authors” or “one of the authors”

-       Lines 165-166: Please clarify what “group one” and “group two” refer to respectively        

I finally suggest a professional revision of the English to eliminate typos and ensure greater comprehensibility

KUB model references (cited in the revision):

Dorigato, L., Philip, G., Bongelli, R., Zuczkowski, A. (2015). Knowing, Unknowing, Believing stances and characters’ dialogic identities in the Harry Potter books. Language and Dialogue, 5(1), 62-89.

Riccioni, I., Bongelli, R., Philip, G., Zuczkowski A. (2018). Dubitative questions and epistemic stance. Lingua, 207, 71-95.

Riccioni, I, Bongelli, R., Zuczkowski, A. (2021). Self-mention and uncertain communication in the British Medical Journal (1840–2007): The decrease of subjectivity uncertainty markers. Open Linguistics, 7(1), 739 – 759.

Riccioni, I., Zuczkowski, A., Burro, R., Bongelli, R. (2022). The Italian epistemic marker mi sa [to me it knows] compared to so [I know], non so [I don’t know], non so se [I don’t know whether], credo [I believe], penso [I think]. Plos One, e0274694

Author Response

We received the report and made an extensive revision of our paper. We are grateful for the comments on the"KUB" model's application to other research domains, such as scientific texts, Italian and English spontaneous conversations, and some types of discourse. We have added the "KUB" Model and Its Applications section, in which we related several studies based on the model. We have revised the details and will reply to the major points listed below:
1. We are sorry for the mistakes. The epistemic stance: "not knowing/uncertain" should be "not knowing whether or believing /uncertain", corresponding to the abbreviated form B.
2.  The terminology: "lexical macro-markers", and "lexical micro-markers" are not used by Zuczkowski et al. However, the two types of markers are under the lexical category. We removed "lexical" in case of misinterpretation. 
The method part:
1.  The participants' information is amplified with the basic socio-demographic features: gender, age, major, and English language proficiency.
2. "The newest version" refers to the "KUB" model in Questions and Epistemic Stance in Contemporary Spoken British English, published by Zuczkowski et al. (2021). The "newest version" is rather ambiguous, so we revised this part, explained the macro-markers and micro-markers, and provided some examples when necessary.
2.  We conducted the literary search under the concordance and imputed keywords, such as "I think", "possibly", "I know", and so on. A total of 12 markers were identified under the knowing/certain position, 17 markers under the believing/uncertain position, and 11 markers under the unknowing/neither certain nor uncertain position. Those words were identified and counted automatically. After that, the two authors marked the morphosyntactic forms separately, sought agreement, and combined the files to check the kappa value. The average kappa value is 0.95, indicating high reliability.
3. The frequencies of markers under each category were counted and used for statistical analysis. 
We enriched the conclusions and added limitations and future development to this section. The discussions and the impact of this study were amplified.
We appreciate your comments and will upload the revised manuscript as soon as possible. 

Reviewer 3 Report

The question of how epistemic stance markers are acquired has seldom been studied within an instructed setting. A study on L2 acquisition of epistemic stance markers by Chinese learners of English is therefore a welcome addition.

The authors propose to analyse the effect of grade and of the type of task (what they call “in-built stylistic message” or “genre-specific context”), in line with studies by Ou & Huang (2016) and Gablasova et al. (2017) which take establish a link between type of discourse (monologic vs dialogic in Gablasova et al.) and use of epistemic stance markers.

The term “grade” is not clearly defined: I assume from reading the Method section (3.1 participants) that it has to do with being in 1st or 2nd year at university. I therefore do not understand why Year 1 students (sophomore) are the experimental group when Year 2 are called “control group”. I would expect a control group to be an equivalent group of English native speaker participants (Year 1 anglophone students?) Moreover, the authors seem to equate “grade” and “proficiency”: having students from a variety of degrees says nothing about their proficiency level in English. No proficiency measure seems to have been provided, nor do we have information about previous exposure to the English language (high school instruction, contact with native input…). Grade alone (ie, being in Year 1 or Year 2) is not likely to have any impact on the use of epistemic stance markers, while proficiency has been shown to correlate with use of modal markers.

The terms “genre-specific context” and “in-built stylistic message” are confusing. It seems to me that the authors proposed oral production tasks, with 4 different discursive genres: narration, description, argumentation, exposition. Contrary to the Gablasova et al. study, the authors do not contrast dialogic vs monologic tasks, as all the proposed tasks are monologic (a form which is less prone to inducing the use of epistemic markers). Another problem is that these tasks require good discursive competence (something which develops with proficiency).

The literature review features relevant studies, but the summary proposed by the authors often lack coherence and are difficult to understand:

Line 45-49: Some features of the KUB models are mentioned but without examples it is very difficult to follow what the authors mean (right side? Left side? What do these notions refer to?) In general, the KUB framework, which seems to be crucial for data analysis, needs to be presented in details, with supporting examples, otherwise data interpretation makes no sense. In particular, it is difficult to understand why “plain declarative sentences” are included in the “morphosyntactic markers” category in Table 1. Micro and macro-markers need to be explained and some examples must be provided for the reader to make sense of the analysis.

Line 56-57 The authors remark that “the pragmatics of spoken communication remains an underresearched area”. I would recommend reading which provides a recent review of L2 pragmatics.

Pérez Vidal, C., & Shively, R. L. (2019). L2 pragmatic development in study abroad settings. In N. Taguchi (Éd.), Handbook of SLA and Pragmatics (p. 355‑371). Routledge.

Line 66-70: the authors discuss Zhang & Sabet (2016), but mention an acronym (IT) 3 lines before explaining it (it refers to “I think”). They explain that the authors “investigate the elasticity of IT” but do not explain what the authors mean by this concept. Their conclusion makes therefore no sense at all: “The result shows that L1 and L2 speakers stretch IT to variable degrees and stop at variable points along the three continua, frequency, position, and cluster.”

Line 89-90 the authors discuss Gablasova & Brezina (2015): the conclusion needs to be rephrased (as such I can’t make sense of it).

Line 111-113 Research questions: the authors mention 2 groups but the two groups haven’t been presented.

Methodology

Regarding the experimental protocol, the authors did not specify whether the participants had some time for planning their response before they started the recording or whether they had to do it with no preparation time. This might considerably change the content of the recordings.

The authors mention line 169 that the recordings were transcribed using Iflytek: how is this app reliable when it comes to transcribing learner data in L2 English? Have the author checked the reliability of the transcription procedure? Have the transcription been manually checked?

Line 169-170 “The authors manually identified the morphosyntactic markers, and make decisions for inclusion.” Based on which criteria?

Data analysis

The authors need to explain what the categories listed in Table 4 and 5 refer to. As such, it is very difficult to understand what markers are analysed.

Line 175-6: This sentence needs reformulation.

Figure 1 & 2: it would be interesting to have the values of the different epistemic markers in the different discursive contexts.

Line 188 some references are missing.

Line 194-6: I don’t see how sentence syntax (independent clause) can be equated with a morphosyntactic marker. To be reformulated.

Line 206-7 some references are missing

Line 217-221 the authors remark that “I think” is likely to occur with hesitation markers and pauses. I totally agree, but when this is the case, many authors consider that “I think” takes a pragmatic value rather than an epistemic one. See the following references for a discussion of the values of “I think”:

Kärkkäinen, E. (2003). Epistemic Stance in English Conversation. John Benjamins Publishing Company.

Mullan, K. (2010). Expressing opinions in French and Australian English discourse : A semantic and interactional analysis. John Benjamins Pub. Co.

Line 223-232: it’s not clear whether the quotes (“because plus the subordinate”) come from verbalizations from the participants or emanate from the authors. It’s not clear either why they think that the relative “that” is added by GS9 to show formality. Please explain.

Examples should be numbered. Many examples illustrate non-epistemic values of ‘I think’: I would recommend making a qualitative analysis of the data, categorizing the different markers under consideration according to their discursive function, before proceeding to the quantitative analysis. As it stands, the paper can’t be considered for publication.

Finally, the paper needs careful proofreading by a native speaker of English.

Author Response

We are glad to receive your review report and revised our manuscript accordingly:

  1. Ou & Huang (2016) and Gablasova et al. (2017)’s studies highly align with our research theme: epistemic stance used in types of discourse (monologic/dialogic), and research on the monologic task type proves insufficient. Providing that research on that topic is limited, we decided to review the previous studies from two aspects: studies on epistemic stance in oral EFL and lay a foundation for introducing the KUB model as the theoretical framework for our research. We added the section: KUB Model and its Applications and addressed the research gap that the model has seldom been applied to EFL instruction.
  2. The experimental group and the control group are equal. We replaced the expression with Group One and Group Two and added the basic demographic information of the two group members. Group one speakers are sophomores with an average level of College English Band Four (CET-4) and received two-year English oral training in college. Group two consists of first-year graduate students averagely leveled with College English Band Six (CET-6) and have been orally trained for four years. CET-4 and CET-6 are the two standardized tests for gauging college students’ English language proficiency in China. The characteristics displayed by the two groups can reflect the purpose and effect of English oral pedagogy and English language proficiency tests in China.
  3. The terms “genre-based questions” and “genre-specific messages” are appropriate and have been substituted for “genre-based context” and “in-built stylistic message”.
  4. The authors rewrote the summaries of the previous studies:
  • The authors delineated the KUB model with examples. The three positions—knowing/certain position, not knowing whether or believing/uncertain position, and unknowing/neither certain nor uncertain position--each having two sides, one evidential (source, access) denoting the left of the slash and the other epistemic (commitment), pointing to the right of the slash.
  • The pragmatics of spoken communication is not an under-researched area, and the remark misaligns with our research purpose: probing into the effect of oral English pedagogy and tests in China.
  • The authors cited Zhang & Sabet (2016)’s study on “I think” (IT) to show that L1 and L2 speakers have different focuses when using IT. Thereby demonstrating the effect of cultural background.
  • We rephrased the conclusions drawn by Gablasova and Brezina (2015).
  • The “two groups’ were mentioned in the previous text in our revised version so that the phrase did not sound abrupt in the research questions.

  1. Methodology
  • They were required to submit the recordings in 3 minutes, and they could have about 30 seconds for preparation after the questions popped up on the screen. However, they were not informed of the content before the test.
  • The recordings were transcribed using the online platform: Iflytek: https://www.iflyrec.com/, which can transcribe the uploaded tape recordings automatically and identify 51 foreign languages, and four dialects. The authors listened to the tape recordings, manually checked the transcribed texts, and sought agreements for some illegible pronunciations.
  • The criteria of the morphosyntactic markers are listed in Tables 1, 2, and 3. The revised version illustrated this part.
  1. Data analysis:
  • Tables 4 and 5 presented the statistical analyses of the overall frequencies of epistemic markers used by Group One and Group Two, respectively.
  • Lines 175-176 were reformed as “Thereby, according to the group, the author built two datasets: the Group One dataset encompassing a total of 36,607 tokens, and the group two dataset consisting of a total of 46,533 tokens.”
  • The independent clause: the plain declarative sentence in the indicative mood (present/past/future) without any lexical marker of epistemicity or evidentiality (Lyons, 1968; Aijmer, 1980) is a type of morphosyntactic marker of the Knowing/Certain position. For example, “When I was a child, I used to play by the lake,” Given that our study is performed in the monologic settings, “I think” is more likely to indicate uncertainty. Still, we did not exclude its pragmatic value.
  • We admitted that “I think” may take a pragmatic value. However, that claim assumes that the epistemic stance is a pragmatic rather than a semantic notion, which usually occurs in conversations (Kärkkäinen 2003; Pouromid 2021; Mullan, 2010).
  • “Because plus subordinate “should be the structure “because that”, which is overused by the participants. “I think that” may be a trunk installed in the speakers’ minds, apart from indicating formality.

We appreciate your comments, and we will upload our revised manuscript soon.

References:

Lyons, J. (1968.) Introduction to theoretical linguistics. London: Cambridge University Press.

Aijmer, K. (1980). “Evidence and the Declarative Sentence.” Acta Universitatis Stockholmiensis. Stockholm Studies in English Stockholm 53: 3-150.

Kärkkäinen, E. (2003). Epistemic Stance in English Conversation. John Benjamins Publishing Company.

Mullan, K. (2010). Expressing opinions in French and Australian English discourse : A semantic and interactional analysis. John Benjamins Pub. Co.

Pouromid, S. (2021) From incompetence to competence: Maintaining intersubjectivity through shifting epistemic stance in intercultural L2 talk in an Asian context, Asian Englishes, (23)2, 166-183.

Round 2

Reviewer 1 Report

The manuscript has improved in several ways. Another round of proofreading will improve the quality of the work further, however.

Author Response

Dear reviewer,

We appreciate your comments and have prebreaded our article in several ways. Please check our revised manuscript.

Reviewer 2 Report

Thank you for reviewing your article. I think the comments raised in my previous review have been addressed adequately.

I only have a few minor remarks:

- Line 8 (Abstract): it would be preferable to cite Zuczkowski et al. (2021) instead of Zuczkowski et al. (2014), since the former contains the definition of the three epistemic positions that the article refers to.

- Line 121: insert a space after (2014).

- Line 123-271: I suggest replacing ‘shew’ with ‘showed’.

- Line 127: insert a space after (2016).

- Line 128: replace ‘unmaking’ with ‘unmasking’.

Tables 1, 2, 3: check type and font size.

- Line 207: delete ‘may not’ (in the KUB model it expresses an ‘impossibility’, and thus Certainty); replace ‘model’ with ‘modal’

- Lines 214-217: there must be a misunderstanding concerning the epistemic future, that is to say, the conjectural use of ‘will’, and the example, consequently, is not appropriate.

I can mention Palmer (1987; Palmer, Frank. 1987. Mood and modality. Cambridge: Cambridge University Press) for a clear example of epistemic future:

((The doorbell is ringing)) That will be the postman

Here the speaker does not refer to something they know will happen in the future (Certainty), but to an assumption on their part (Uncertainty), based on prior knowledge (concerning, for example, the fact that the postman usually arrives at that time and if he has to deliver mail he rings the bell).

“Epistemic future arises when future expressions are used with lower present or past tenses without making a prediction. This should not happen if future expressions were simply future tenses.” Giannakidou, A., & Mari, A. (2018). A unified analysis of the future as epistemic modality. Natural Language & Linguistic Theory, 36(1), 85-129.

- Line 256: groups’ instead of group’s?

- Line 492: check type and font size.

References: check the references carefully according to the journal guidelines, because sometimes the authors are cited by their first name and sometimes only by their first initial.

Author Response

Dear reviewer,

We appreciate your comments and revised our manuscript accordingly:

  1. Zuczkowski et al. (2014) were replaced with Zuczkowski et al. (2021) to better demonstrate the three epistemic stances.
  2. Line 121-207. We made minor revisions in terms of space, expressions, and style.
  3. Thanks for the advice on the epistemic future, and we changed the example and added the recommended one: “(The doorbell is ringing.) That will be the postman” can be paraphrased as “I (now) hear the doorbell ringing and make an assumption (Uncertainty) that the postman is at the door”, based on prior knowledge (concerning, for example, the fact that the postman usually arrives at that time and if he has to deliver mail, he rings the bell) (Palmer, 2001).
  4. We reformatted the references according to the journal guidelines.

Reference:

Palmer, F. (2001). Mood and modality. Cambridge: Cambridge University Press

Reviewer 3 Report

Many thanks to the authors for the time they took to answer my comments. The revised version features a clearer presentation of the KUB model and more detailed information about the participants, including their proficiency level. The authors reply adequately to my comments.

A clear formulation of the research question and a clearer explanation of the choice of task (why choose a monologic task? Why would you expect epistemic stance development between year 1 and 4?) would help the reader navigate the paper. 

But extensive English language editing is needed to make this paper publishable.

Author Response

Dear reviewer.

We appreciate your comments and added the recommended details.

  1. We chose the monologic tasks due to the following reasons:

     Research on the effect of epistemic stance on ESL (English as a Second Language) learners’ verbal presentations in monologic tasks has proved insufficient. (eg. Gablasova et al., 2017; Ou & Huang, 2016).

    Designing the monologic tasks aligns with our research purpose: testing the effect of grade, which reflects the sense of College English training in China, and the effect of genre-based questions. We need to control the potential affective variables to explore the effect of the two independent variants. The effect of language proficiency may be mitigated by learners involved in interactions, but in monologic tasks, learners are left to their own resources without being affected by their interlocutors (Robinson, 2011). The monologic tasks are optimal since we can elicit unspoiled data.

   The monologic tasks are easy to operate, and the data collected is easy to process. As a result, the research outcomes are of high reliability.

  1. Perry (1970) recognized that epistemic development continues long beyond childhood and notified nine positions regarding college students’ epistemic growth. Many previous studies proved that there is indeed an epistemic development between sophomores and seniors in college ((Widick et al., 1975; Meyer, 1977; Clinchy et al., 1977; Chandler et al., 2002). Therefore, the epistemic stance discrepancy is guaranteed by selecting sophomores and first-year graduates as research subjects.

References:

Perry, W. G. (1970). Forms of Intellectual and Ethical Development in the College Years. Holt, Rinehart and Winston.

Meyer, J. P. (1977). Intellectual Development: Analysis of Religious Content. The Counseling Psychologist, 6(4), 47–50.

Chandler, M.J., Hallett, D., & Sokol, B.S. (2002). Competing claims about competing knowledge claims. In Personal epistemology: The psychology of beliefs about knowledge and knowing (pp. 145–168). Erlbaum.

Clinchy, B., Judy, L., & Young, P. (1977). Epistemological and Moral Development in Girls from a Traditional and a Progressive High School. Journal of Educational Psychology, 69(4), 337–343.

Widick, C., Knefelkamp, L.., & Clyde A. P. (1975). The Counselor as a Developmental Instructor. Counselor Education and Supermiion, 14, 286–296.

Robinson, P. (2011). Task-Based Language Learning. Amsterdam: John Wiley & Sons.